# Development and Characterization of a Luciferase Labeled, Syngeneic Murine Model of Ovarian Cancer

**DOI:** 10.3390/cancers14174219

**Published:** 2022-08-30

**Authors:** Shonagh Russell, Felicia Lim, Pamela N. Peters, Suzanne E. Wardell, Regina Whitaker, Ching-Yi Chang, Rebecca A. Previs, Donald P. McDonnell

**Affiliations:** 1Department of Pharmacology and Cancer Biology, Duke University School of Medicine, Durham, NC 27710, USA; 2Department of Obstetrics and Gynecology, Division of Gynecologic Oncology, Duke University School of Medicine, Durham, NC 27710, USA

**Keywords:** ovarian cancer, immune profiling, immunotherapy, intraperitoneal, ovarian intrabursal, murine, high-grade serous, syngeneic, STOSE

## Abstract

**Simple Summary:**

We developed a new labeled mouse cell line to study ovarian cancer. The STOSE mouse cell line was engineered to express a luciferase label to enable real-time tumor monitoring by imaging. We characterized the growth of the cell line in mice and identified the immune cells within the formed tumors. We treated the mice with immunotherapy, which had no effect on tumor growth. Overall, the STOSE.M1 luc model closely resembles ovarian cancer in humans and will further aid in our understanding and treatment of this disease.

**Abstract:**

Despite advances in surgery and targeted therapies, the prognosis for women with high-grade serous ovarian cancer remains poor. Moreover, unlike other cancers, immunotherapy has minimally impacted outcomes in patients with ovarian cancer. Progress in this regard has been hindered by the lack of relevant syngeneic ovarian cancer models to study tumor immunity and evaluate immunotherapies. To address this problem, we developed a luciferase labeled murine model of high-grade serous ovarian cancer, STOSE.M1 luc. We defined its growth characteristics, immune cell repertoire, and response to anti PD-L1 immunotherapy. As with human ovarian cancer, we demonstrated that this model is poorly sensitive to immune checkpoint modulators. By developing the STOSE.M1 luc model, it will be possible to probe the mechanisms underlying resistance to immunotherapies and evaluate new therapeutic approaches to treat ovarian cancer.

## 1. Introduction

Approximately 90% of all ovarian cancers are of epithelial cell origin [1], with the high-grade serous ovarian cancer (HGSOC) histological subtype being the most common and that which has the poorest outcomes [2]. The primary interventions for advanced-stage HGSOC in the frontline setting include surgery and neoadjuvant or adjuvant platinum and taxane-based chemotherapies [3]. Anti-vascular endothelial growth factor (VEGF) therapy and poly (ADP-ribose) polymerase (PARP) inhibitors have emerged as effective maintenance therapies [4,5,6,7,8]. Additionally, VEGF therapy [9] is also used in the frontline setting for some patients. Although most patients with HGSOC initially respond well to these therapies, recurrence and development of resistance impede long-lived remissions [10]. Unfortunately, for patients who progress on existing frontline therapies there are few effective therapeutic options and the available drugs rarely lead to durable responses. Thus, there is an unmet need to develop interventions that can be used as single agents or combined with existing therapies to reduce recurrence and treat drug-resistant disease. Recently, new targetable vulnerabilities have been identified in HGSOC [11,12], and therapies targeting these are currently being evaluated for clinical efficacy. However, these new drugs/treatment regimens will likely result in only incremental improvements in outcomes.

Given the successes of contemporary immunotherapies in some solid tumors, including melanoma [13,14], non-small cell lung cancer (NSCLC), and kidney cancer [15,16], it is disappointing that these drugs are only effective in a small number of patients with HGSOC. Approved immunotherapies for ovarian cancer are based on tumor agnostic biomarkers. Dostarlimab, an anti PD-1 immunotherapy, is approved for patients with recurrent solid cancers, including ovarian cancer, whose tumors have a DNA mismatch repair deficiency (dMMR). Pembrolizumab, a second anti PD-1 therapy, is approved for patients with recurrent cancer, including patients with HGSOC; if the patient’s tumor has dMMR, a high tumor mutation burden (defined as ten mutations/megabase), or exhibits microsatellite instability. Studies of single-agent checkpoint blockade in unselected patients with HGSOC have shown minimal efficacy [17,18].

Frontline trials in which checkpoint blockade was combined with chemotherapy and followed by standard of care maintenance therapy have also failed to meet their primary endpoints [19,20,21]. As with other solid tumors that respond poorly to immunotherapy, it is generally believed that epithelial ovarian tumors, and platinum-resistant HGSOC tumors, in particular, are immunologically cold [22] with a low neoantigen burden and/or an immunosuppressive tumor microenvironment [23]. Therefore, research priorities include understanding the molecular basis of resistance and developing approaches to increase ovarian tumor immunity, particularly in HGSOC. Furthermore, studies exploring new approaches to favorably manipulate tumor-associated immune cell activity and increase the clinical utility of existing immunotherapies will significantly benefit this disease.

Notwithstanding the potential differences in the biology of HGSOC in humans and mice, the lack of tractable syngeneic murine models of this disease has significantly limited progress. In this study, we have addressed this issue by developing and characterizing a new model (STOSE.M1 luc) of HGSOC. Specifically, luciferase was stably expressed in STOSE cells (originally generated by the spontaneous transformation of TP53 wild-type ovarian surface epithelial (OSE) cells, from Friend leukemia virus B (FVB) mice) [24]. This engineered cell line was then serially passaged in the ovaries of immunocompetent mice to generate the stable STOSE.M1 luc model. We characterized this model extensively for its ability to grow as a primary tumor and metastasize and evaluated the repertoire and function of immune cells in tumors. This new, well-characterized model, which we will make available, will be a valuable tool for studying the biology and pharmacology of therapies in HGSOC.

## 2. Materials and Methods

### 2.1. Cell Culture

STOSE cells were cultured in RPMI-1640 media (ThermoFisher Scientific, Waltham, MA, USA), supplemented with 8% fetal bovine serum (FBS, Sigma, St. Louis, MO, USA), 1mM sodium pyruvate (Invitrogen, Waltham, MA, USA), and 0.1mM nonessential amino acids (Invitrogen) at 37 °C and supplied with 5% CO_2_. STOSE cells, a murine ovarian cancer cell line derived from the ovarian surface epithelium of FVB mice, were a kind gift from Dr. Barbara Vanderhyden (University of Ottawa). Cell lines were tested using a PCR-based assay and confirmed to be negative for mycoplasma. Before implantation in mice, STOSE cells underwent Duke Mouse Impact III cell line screening for pathogens and were negative.

### 2.2. Luciferase Labeling of the STOSE Cell Line

Human embryonic kidney (HEK) 293T cells were cultured in 10 cm plates containing DMEM (ThermoFisher Scientific) + 10% FBS to 30–40% confluence. Transfection reagents: 263 µL Opti-MEM (31985062, ThermoFisher Scientific), 17 µL of FuGENE-6 (E2692, Promega, Madison, WI, USA), 2.8 µg of lenti-luc-p2a-neo (Addgene 105621), 2.8 µg of psPAX2 (Addgene 12260) and 0.28 µg of pCMV-VSVG (Addgene 8454) were mixed and incubated for 30 min prior to adding dropwise to HEK 293T cells followed by incubation at 37 °C overnight to generate lentivirus. Lenti-luciferase-P2A-Neo was a gift from Christopher Vakoc (Addgene plasmid # 105621; Accessed on 5 August 2022—http://n2t.net/addgene:105621; RRID:Addgene_105621 [25], psPAX2 was a gift from Didier Trono (Addgene plasmid # 12260; Accessed on 5 August 2022—http://n2t.net/addgene:12260; RRID:Addgene_12260), and pCMV-VSV-G was a gift from Bob Weinberg (Addgene plasmid # 8454; Accessed on 5 August 2022—http://n2t.net/addgene:8454; RRID:Addgene_8454) [26]. After 16–18 h, media were replaced with fresh DMEM + 30% FBS. Virus-containing media were harvested every 24 h for 48 h and filtered through a 0.45 µM filter membrane. A total of 10mL of virus plus 4 µg/mL polybrene was added to a 10 cm dish containing STOSE cells (P12), 50% confluent, and incubated overnight at 37 °C. Following viral transduction, luciferase labeled STOSE cells were selected in 750 µg/mL G418, media were replenished, and cells split as needed over ten days. Luciferase expression was confirmed by luciferase plate assay following the instruction of Promega luciferase assay protocol using a non-commercial luciferase buffer as described by Dyer et al., 2000 [27].

### 2.3. Mouse Studies

All procedures were approved by the Duke University Institutional Animal Care and Use Committee (IACUC) under protocols A108-20-05 and A175-19-08.

#### 2.3.1. Pilot Study of STOSE-luc Cells in Mice

FVB female mice were obtained from an in-house breeding colony at the Duke Cancer Institute. Breeders were originally purchased from The Jackson Laboratory. Mice were maintained under specific pathogen-free, temperature- and humidity-controlled conditions, with a 12 h light/12 h dark schedule. During tumor studies, activity level, coat condition, and changes in behavior, such as guarding and hiding, were monitored daily. Mouse weight was assessed twice weekly, and tumor growth was assessed weekly by quantifying the average luminescence intensity by In Vivo Imaging System (IVIS) Lumina XR.

A pilot study was performed to determine the optimal cell number for ovarian tumor formation by ovarian intrabursal injection. Ten female FVB mice (6–7 weeks old) were implanted with 1 × 10^5^–1 × 10^6^ STOSE-luc cells. Group 1 (*n* = 5) mice had 1 × 10^5^ cells implanted into the left ovary and 1 × 10^6^ cells implanted into the right ovary. Group 2 (*n* = 5) had 2.5 × 10^5^ cells implanted into the left ovary and 5 × 10^5^ cells implanted into the right ovary. Cell implantation was into the ovarian bursa under sterile surgical conditions.

Surgery was performed as follows and was the same for all ovarian intrabursal in vivo studies. Mice were anesthetized in an inhalation chamber (2% Isoflurane, 4% oxygen) and maintained via a nose cone (1% Isoflurane, 4% oxygen) throughout the surgical process on a warming blanket. Prior to surgery, mice received a 5 mg/kg dose of carprofen subcutaneously (SC). The dorsal area below the ribs was shaved with an electronic razor and the skin was sterilized with betadine and alcohol. A 0.5 cm horizontal incision was made through the skin above the ovarian fat pad, followed by a vertical incision through the abdominal muscle wall. The ovary was externalized and 5 µL volume of cell suspension (luciferase labeled STOSE cells, STOSE- luc, in PBS) was injected into the ovarian bursa using a Hamilton syringe. The ovary was returned into the abdomen and abdominal wall layers were reapproximated and sutured. One drop of bupivacaine (0.25%) was added on top of the incision site. A wound clip was placed on the incision site. This was repeated for the contralateral ovary. The mouse was then removed from anesthesia and kept in a clean cage and monitored until conscious. Post-surgery, mice were administered a 5 mg/kg dose of carprofen subcutaneously daily for the following 48 h. The mice were monitored for recovery for ten days and the wound clips were removed when the incision healed.

#### 2.3.2. Repassage STOSE.M1 luc Cells in Mice

Tumors from the STOSE.luc pilot study were harvested, dissociated, and combined into a single cell suspension under sterile conditions and re-implanted, as described above, in five new FVB female mice (6–7 weeks of age) into the right ovary. Due to the small number of cells harvested from these tumors, the cell number injected was unknown. The remaining cells after implantation were cultured under normal tissue culture conditions, expanded and frozen down in 10% DMSO for future studies, named STOSE.M1 luc. The STOSE.M1 luc tumors that developed were harvested at the study end, combined into a single cell suspension, cultured under normal tissue culture conditions, expanded and frozen down in 10% DMSO for future studies, named STOSE.M2 luc.

#### 2.3.3. Pilot Cell Number Study of STOSE.M1 luc

A pilot study was performed to determine the optimal cell number for ovarian tumor formation from ovarian intrabursal injection of the STOSE.M1 luc cell line. Ten female FVB mice 6–7 weeks old were implanted with 1 × 10^4^–1 × 10^5^ STOSE.M1 luc cells. Group 1 (*n* = 5) mice had 1 × 10^4^ cells implanted into the left ovary and 1 × 10^5^ cells implanted into the right ovary. Group 2 (*n* = 5) had 2.5 × 10^4^ cells implanted into the left ovary and 5 × 10^4^ cells implanted into the right ovary. Cell implantation was into the ovarian bursa under sterile surgical conditions described above, and tumor growth was monitored weekly by IVIS imaging. At the completion of the study, tumors were dissociated into single cell suspension for immune profiling, described below.

#### 2.3.4. Intraperitoneal STOSE.M1 luc Growth Study

STOSE.M1 luc cells, 1 × 10^6^ in 200 µL of Hank’s Balanced Salt Solution (HBSS), were injected intraperitoneally (*n* = 5 female FVB mice 6–7 weeks of age). Tumor growth was monitored by IVIS imaging as described below.

#### 2.3.5. Subcutaneous STOSE.M1 luc Growth Study

STOSE.M1 luc cells, 1 × 10^6^ in 200 µL HBSS, were injected subcutaneously into the left and right flank (*n* = 5 female FVB mice 6–7 weeks of age). Tumor growth was monitored by IVIS imaging and caliper measurement.

### 2.4. IVIS Imaging

Growth of STOSE- luc and STOSE.M1 luc in ovarian bursa (intrabursally), intraperitoneally, and subcutaneously was monitored by IVIS Lumina XR imaging system 1–2 times per week. To image tumors, mice were injected intraperitoneally with 100 µL of D-luciferin sodium salt (Regis Technologies 103404-75-7) reconstituted in PBS (15.1 mg/mL). Five minutes post luciferase injection, mice were anesthetized in an inhalation chamber (3% Isoflurane, 4% oxygen) and maintained in half the dose of isoflurane (1.5%) via nose cone throughout the imaging procedure on a 37 °C stage. Images were taken using exposure times of 1, 30, 60 and 180 s and region of interest (ROI) drawn for each tumor or metastatic area and calculated as average radiance [p/sec/cm^2^/sr] or total flux [p/s]. Images from each study were normalized to the same color range.

### 2.5. Tumor Dissociation to Generate Single Cell Suspension

Tumors were isolated, minced on a petri dish in media (DMEM with 5% FBS) and digested enzymatically with 100 µg/mL DNase I (D5025-150KU, Sigma-Aldrich) and 1 mg/mL collagenase (Collagenase A, Cat 10103586001, Sigma–Aldrich, St Louis, MO, USA), then shaken (250 rpm) at 37 °C for 45 min to 1 h. The resulting cell slurry was filtered through a 40 µM strainer (Cat 431750, Corning, Corning, NY, USA) to produce single cell suspensions. The digestive enzymes were diluted by adding media and then cells spun down to remove media. Red blood cells were lysed with ammonium chloride potassium (ACK) lysis buffer (Cat A1049201, ThermoFisher Scientific, Waltham, MA, USA) for 4 min at room temperature. Following red blood cell lysis, cells were washed with PBS before counting on a hemocytometer using trypan blue and then stained for flow cytometry staining, expanded in cell culture, or re-implanted in mice. To dissociate the metastases, metastases visible by eye were excised from the peritoneum and all organs contained within the peritoneal cavity. The metastases were then pooled and processed as described above.

### 2.6. Flow Cytometry of Tumor and Immune Cells

Single cell suspensions (1 × 10^6^ cells in 50 µL) generated as described above were incubated with Live/dead cell stain (see Appendix A) in PBS for 10 min at 4 °C. Cells were spun down at 2000 rpm and incubated with anti-CD16/32 (Catalog 14-0161-85, ThermoFisher Scientific) in flow buffer (10 g BSA and 0.2 g NaN3 in 1L PBS) for 15 min at 4 °C. Following this, cells were stained with an extracellular antibody cocktail in Brilliant Stain Buffer (Cat 566349, ThermoFisher Scientific) for 30 min at 4 °C. The antibodies used are listed in Appendix A. For intracellular staining, cells were fixed and permeabilized using 50 µL of Fix Perm solution from eBioscience Foxp3 Transcription Factor Staining Buffer Set (Cat 00-5523-00, ThermoFisher Scientific) for 30 min at 4 °C. Cells were spun down and intracellularly stained with desired antibody prepared in Perm buffer for 30 min at 4 °C. Cells were then fixed in fix buffer for 30 min at 4 °C, spun down and re-suspended in 200 µL in flow buffer for flow cytometry. Multicolor flow cytometry was performed in BD Fortessa 16 color analyzer. The flow cytometry data were analyzed by FlowJo V10 software (FlowJo, LLC, Vancouver, BC, Canada). To identify immune cell populations and changes between the ovarian intrabursal model cancer sites (primary tumor, ascites, and metastases), we utilized Uniform Manifold Approximation and Projection (UMAP) for dimension reduction and FlowSOM plugins for population clustering in the FlowJo software. Before running UMAP, the cells in each sample were downsampled to a smaller number of cells to enable data processing. In most metastatic samples, all cells were included in the down sampling due to the low number. After downsampling, each sample was labeled with a keyword (1—primary, 2—ascites, 3—metastasis) to enable deconvolution post concatenation. The samples from each site were concatenated together, followed by the concatenation of the site files. UMAP algorithm was run on the concatenated file to identify clusters while maintaining global structure, then the FlowSOM plugin was utilized to visualize the population clusters and identify the delineating markers. From the concatenated file, each site was pulled out using the keyword, and the individual FlowSOM clusters were overlaid onto the individual site UMAP to show differences in the same populations between sites.

### 2.7. Anti-PD L1 Treatment of STOSE.M1 luc Tumors

#### 2.7.1. Intraperitoneal

STOSE.M1 luc cells were injected intraperitoneally, 1 × 10^6^ cells in 200 µL HBSS, as described for the pilot study. Once tumors were established, as observed by IVIS nine days post injection, mice were randomized into groups, *n* = 8 for PBS control, and *n* = 4 for anti PD- L1 (Cat BE0101, Bio XCell, Nusajaya, Malaysia). PBS group was injected with 100 µL PBS intraperitoneally on days 12 and 15 and the anti PD-L1 group was injected with 200 µg of anti PD-L1 in 100 µL PBS intraperitoneally on days 15, 18, 22, and 26. Tumor growth was monitored weekly by IVIS imaging. Mice were euthanized on day 28 and necropsy was performed. Tumors were then dissociated for immune profiling.

#### 2.7.2. Subcutaneous

STOSE.M1 luc cells were injected subcutaneously, 1 × 10^6^ cells in 200 µL HBSS per flank, as described for the pilot study. Once palpable, tumors were measured by calipers until 50 mm^2^ (Day 18 post-injection), then mice were randomized into groups, *n* = 10 for PBS control, and *n* = 5 for anti PD-L1 (Cat BE0101, Bio XCell). PBS group was injected with 100 µL PBS intratumorally in the largest flank tumor (D19, D22) and the anti PD-L1 group was injected with 200 µg of anti PD-L1 in 100 μL PBS intraperitoneally (D22, D26, D29, D33). Tumor growth was monitored weekly by IVIS imaging. Mice were euthanized on day 35 and necropsy performed. Tumors were then dissociated for immune profiling.

### 2.8. Statistical Analyses

Experiments were performed with a minimum of three biological replicates for each experimental group. For all statistical analyses, GraphPad Prism software was used. Statistical tests utilized included unpaired t-test, Mann–Whitney test, and Kruskal–Wallis test followed by Dunn’s multiple comparisons test, two-way ANOVA followed by Tukey’s post hoc test or Šidák’s multiple comparisons test. Statistical significance was defined as *p* < 0.05.

## 3. Results

### 3.1. Development and Characterization of the Syngeneic STOSE.M1 luc HGSOC Model

The STOSE mouse cell line models human HGSOC and has been propagated as tumors in syngeneic hosts [24]. However, palpation of orthotopic ovarian tumors allows only a semi-quantitative assessment of tumor growth, and real-time measurements of metastasis are not possible. Therefore, there is a need for a tractable syngeneic model to quantitatively track tumor progression. To enable time-resolved, quantitative assessments of tumor burden, we developed the STOSE-luc cell line, a luciferase-labeled version of the well-characterized STOSE cell line. To this end, STOSE cells were stably transduced with lentivirus expressing luciferase. The tumorigenicity of the resulting cells was assessed in vivo following ovarian intrabursal injection of 1 × 10^5^ to 1 × 10^6^ cells into ten mice. Successful implantation of the STOSE-luc cells in the mouse ovaries was confirmed by IVIS imaging (Appendix A), and after approximately four months, tumors formed in three mice (Appendix A). The resultant tumors were harvested, and the cells, termed STOSE.M1 luc, were derived from the dissociated tumors (Appendix A). These cells were re-implanted into mice, and tumorigenicity was confirmed by IVIS imaging after 28 days (Appendix A). By study end on day 55, mice developed primary tumors (Appendix A), ascites fluid in the peritoneal cavity (Appendix A), and metastases (Appendix A). These metastatic sites included the peritoneum, contralateral un-implanted ovary, adnexa, uterus, bowel, liver, ureter, mesentery, omentum, spleen, kidney, diaphragm, and lungs.

The growth of the STOSE.M1 luc cell line implanted in either the ovarian bursa, intraperitoneally, or subcutaneously was also evaluated. Implantation of as few as 1 × 10^4^ cells into the ovarian bursa resulted in the formation of approximately 0.5 g primary tumors and metastases (with ascites) after five weeks (Figure 1A–C). Implantation of 1 × 10^6^ cells intraperitoneally or subcutaneously resulted in intraperitoneal (Figure 1D–F) or subcutaneous (Figure 1G–I) tumor formation, respectively. However, the subcutaneous model developed less ascites and fewer metastases when compared to the intraperitoneal or intrabursal models. Overall, the growth characteristics of this immunocompetent model (Table 1 afford the possibility of assessing the impact of therapeutics/manipulations on HGSOC in real-time.

### 3.2. Lymphoid Cell Infiltration at Sites Where STOSE.M1 luc Cells Reside during Tumor Progression

Intratumoral lymphoid cell infiltration is prognostic for improved outcomes in multiple cancers, including ovarian cancer [28,29,30]. However, the lack of validated syngeneic ovarian cancer models has made it challenging to study the biology of lymphoid cells in the context of tumors. To address this issue, we evaluated the lymphocyte cell repertoire/functionality in immune cells isolated from the primary tumor, ascites, and metastases in the STOSE.M1 luc model. To survey the global changes in lymphoid markers on the CD45^+^ immune cells and to identify phenotypic changes in tumor associated immune populations not accessible using traditional gating strategies, we performed uniform manifold projection analysis (UMAP) combined with FlowSOM clustering (Appendix A). In this manner, it was determined that the primary tumor, ascites, and metastases each harbored lymphoid cells but that the populations (denoted by individual colors) and the number of cells contained within individual populations differed between sites (Figure 2A–C and Appendix A).

Compared to other sites, and even after considering tumor weight as a variable, we observed reduced immune cell and lymphocyte infiltration in metastases (Appendix A). Lymphocyte infiltration was further quantified using traditional gating strategies to profile B and T cell subsets (Appendix A). It was determined in this manner that B cells (Figure 2D,E) were the predominant lymphoid cell, with T cells comprising less than ten percent of all immune cells present (Figure 2D). The infiltration of memory B cells was reduced in metastases compared to the primary tumor (Figure 2F). MHCII was not expressed equivalently in B cells indicating reduced antigen presentation, likely resulting in impaired T cell activation in the ovarian tumor microenvironment (Figure 2G) .

Analysis of the T cell subtypes revealed that CD4^+^ T cell infiltration was highly variable (Figure 2H–L) and that CD8^+^ T cells comprised approximately 5% of CD3^+^ lymphocytes in the primary tumor, ascites, and metastases (Figure 2K). Gamma delta^+^ T cells are the main T-cell subtype identified in the tumors from patients with HGSOC [31] and are likewise dominant in this model (Figure 2L). The expression of T cell activation markers was also evaluated to assess T cell functionality. Notable was the observation that CD27, a co-stimulatory marker involved in T cell activation and survival, was expressed in more CD4^+^ and CD8^+^ T cells isolated from metastases compared to primary tumors (Figure 2M–O). In contrast, ICOS expressing CD4^+^ and gamma delta T cells were decreased in metastases compared to primary tumors (Figure 2M–O). Intratumoral CD4^+^ and CD8^+^ T cells expressed varying levels of the activation markers, CD44 and CD69 (Appendix A).

### 3.3. T Cell Subsets and Functional Markers Are Altered between Tumor Sites

T cell subsets, including T helper (Th), T regulatory (T reg), and memory T cells, can be pro or anti-tumorigenic. Therefore, we sought to determine the function of these cells in this model by analyzing immune cells isolated from the primary tumor, ascites, and metastases. The T reg and T helper markers on CD4^+^ CD8- T cells were first assessed (Figure 3A and Appendix A). T reg cells were present in primary tumors and metastases, likely a reflection of the immunosuppressive environment in HGSOC (Figure 3B). However, T reg and T helper cells were absent in the ascites (Figure 3A). Th1 cells were absent in all sites of this model, and Th2 was the predominant population in metastases (Figure 3C). Conversely, Th2 and Th17 cells were present in the primary tumor at similar levels (Figure 3C,D). Some CD4^+^ T cells expressed multiple T helper markers, making it difficult to assign them to a distinct T helper subset (data not shown).

The expression of markers associated with T helper function were also assessed. Th17 cells expressed more functional markers than Th2 cells (Figure 3E,F). Although most Th2 cells expressed granzyme B, they also expressed the inhibitory receptor PD-1, suggesting that the antitumor activity of these cells was likely inhibited. The Th17 subsets lacked expression of PD-1, and in metastases, Th17 cells exhibited broad expression of functional markers (Figure 3F).

Memory T cells are engaged to maintain tumor regression and are required for sustained response to immunotherapies [32,33,34]. Thus, the infiltration of effector, central, resident, and peripheral CD4^+^ and CD8^+^ memory T cells [35] was assessed (Appendix A). CD4^+^ memory T cells were present in the primary tumor, ascites, and metastases (Figure 3G–I), whereas CD8^+^ memory T cells were only present in primary tumors and metastases (Figure 3J–M). The number of central memory T cells was extremely low in this model (Figure 3G,J), with effector memory cells being the predominant T memory subset (Figure 3H,K). CD4^+^ resident memory cells were predominantly found in the ascites (Figure 3I), whereas CD8^+^ resident memory cells were only present in the primary tumor (Figure 3L). CD8^+^ peripheral memory T cells were only present in the primary tumor (Figure 3M). A recent study identified four distinct CD8^+^ memory populations using CD62L and CX3CR1 markers [36]. In this model, we identified two populations, CD62L${}^{-}$ CX3CR1^+^ and CD62L${}^{-}$ CX3CR1${}^{-}$, in the primary tumor, ascites, and metastases (Figure 3N). CD62L${}^{-}$ CX3CR1^+^ memory cells, which are more cytotoxic but less proliferative, and CD62L${}^{-}$ CX3CR1${}^{-}$ memory cells, which are less cytotoxic and more proliferative, were increased and decreased in metastases, respectively.

### 3.4. Pro and Anti-Tumorigenic Myeloid Cells Are Present in All STOSE.M1 luc Tumor Sites

Myeloid cells can be pro- or anti-tumorigenic, for example, intratumoral myeloid-derived suppressor cells (MDSCs) are associated with decreased survival in patients with ovarian cancer [37]. Thus, we assessed the myeloid cell repertoire/functionality utilizing UMAP and FlowSOM analysis in the STOSE.M1 luc model (Figure 4A–C and Appendix A). These analyses highlighted multiple myeloid populations that differ between the primary tumor, ascites, and metastases and established that there were fewer myeloid cells in the metastases. Further characterization revealed that the myeloid cells were predominant over lymphoid cells in the primary tumors and ascites (Figure 4D and Appendix A). Unexpectedly, given their positive role in immunity, M1 polarized macrophages dominated over those that were polarized towards an M2 phenotype in primary tumors (Figure 4E,F). In metastases, the ratio of M1:M2 macrophages was reduced and more reflective of an immunosuppressed microenvironment. We further assessed macrophage function by measuring the expression of CD80 a co-stimulatory molecule used for T cell activation. The majority of M1 macrophages expressed CD80. However, CD80 expression, which is most commonly reduced on M2 macrophages, was expressed on the majority of M2 cells in the ascites (Figure 4G).

MDSCs are myeloid cells that can be stratified into granulocytic/polymorphonuclear (G- or PMN-MDSCs) and monocytic (M-MDSCs). Both subtypes of MDSCs were found in the STOSE.M1 model (Figure 4H,I), with M-MDSCs being predominate in all three sites and enriched in the ascites.

Finally, in our evaluation of myeloid cells, we evaluated the dendritic cell (DC) repertoire (Appendix A), which comprises a small percentage of tumor infiltrating immune cells in our model (Figure 4J). Most DCs present were classified as “mature” expressing both MHCII and CD80 [38,39], as well as functional markers ICOS-L and OX40L (Figure 4K). OX40L was significantly reduced in metastasis compared to the ascites (Figure 4L) .The representation of the three major DC subsets, conventional (cDC1, cDC2) and plasmacytoid (pDC) was also examined, revealing, as is the case in most tumors, that the cDC2 subset was dominant (Figure 4M–O).

### 3.5. Subcutaneous and Intraperitoneal STOSE.M1 luc Models Are Poorly Responsive to Anti PD-L1 Immunotherapy

In general, ovarian cancers do not respond well to immunotherapy, and there is an unmet need for animal models that can be used to understand this biology and evaluate new therapeutic modalities. Thus, we evaluated the efficacy of anti PD-L1 immunotherapy on the STOSE.M1 luc model in vivo. We found that the drug was without effect in the IP (Figure 5A–C) and SC models (Figure 5D–F) when assessed using IVIS or by measuring post necropsy tumor weight. We also performed immune cell profiling of the tumors in these studies. In the IP model, T cell infiltration was unchanged (Figure 5G,H), but some T cell subset markers did change (Figure 5I–L). Specifically, on CD8^+^ T cells, the inflammatory cytokine IFN-Gamma increased (Figure 5I), and the inhibitory marker TIM3 decreased (Figure 5J) with anti-PDL1 treatment. However, granzyme B and PD-1 did not change (Figure 5K–L). The most notable changes observed in the SC model were that gamma delta T cell infiltration decreased (Figure 5M), and the inhibitory receptor PD-1 increased in CD8^+^ T cells (Figure 5M–R) with anti PD-L1 treatment. The importance of these changes remains to be determined.

## 4. Discussion

There is an urgent need for syngeneic models of ovarian cancer as few currently available models reflect the known biology of the disease. Further, few existing models are labeled in a manner that allows real-time assessments of tumor growth and metastasis. The immunocompetent STOSE.M1 luc model was developed to address these shortcomings and recapitulates many facets of human HGSOC, albeit without a TP53 mutation. One limitation of the STOSE model is the cells were derived from ovarian surface epithelium (OSE) cells, and it is a widely held opinion that fallopian tube epithelium (FTE) is the origin of ovarian cancer [40,41]. However, it has been established that mutations in FTE or OSE cells can lead to metastatic HGSOC [42], necessitating the development of models derived from both cell types. Another benefit of this model is that it is derived from FVB mice. FVB is a different genetic background than C57BL/6J from which the common ovarian cancer models, ID-8 and IG-10, were derived. The different mouse strains allow hypothesis testing with confidence that any results are ovarian cancer specific rather than mouse strain specific.

One significant practical advantage of the STOSE.M1 model is the relatively short time it takes to achieve useful tumor growth endpoints following ovarian intrabursal injection (23 to 44) days or intraperitoneal injection (12 to 33 days). This is considerably faster than the time required for tumor growth in the ID-8 and IG-10 models [43,44]. Intrabursal models are the gold standard for mouse ovarian cancer studies but require more time and specialized technical expertise. Therefore, intraperitoneal (IP) models are more commonly used to bypass those challenges, with subcutaneous (SC) models also having some utility [45,46]. Importantly, we have shown that the STOSE.M1 model forms tumors after IP or SC injection but that SC tumors grow slower with unpredictable metastatic spread. Tumors in the peritoneal cavity undergo mechanical sloughing, which contributes to metastasis [47,48,49], whereas this does not occur in the SC model limiting intraperitoneal metastasis. We have shown that IP injection of STOSE.M1 cells leads to intraperitoneal tumor development highlighting its likely utility in studying ovarian cancer biology and drug efficacy.

One caveat of labeled cell models, such as those expressing luciferase, green fluorescent protein (GFP), or red fluorescent protein (RFP), is that the marker can in and of itself be immunogenic [50,51]. However, by repeated passaging of cells through immunocompetent mice, we have selected a subline of luciferase expressing cells that are not cleared by the immune system. This is likely due to the selected cells having a particularly low level of luciferase expression compared to the initial STOSE-luc, which we have observed in other models we have established.

We developed several myeloid and lymphoid panels to profile the immune cell repertoire and function in our model of ovarian cancer. In human ovarian cancer, the abundance of tumor infiltrating lymphocytes (TIL) [52,53,54], T Regs [55], M-MDSCs [37], and natural killer (NK) cells [56] have been shown to be clinically meaningful. Indeed, ovarian tumors are generally considered immunologically cold and immunosuppressive [57], a state echoed in the STOSE.M1 model, where tumors were infiltrated with MDSCs and T regs. In the lymphoid compartment, T cells expressed low granzyme B and the majority of these cells did not express activation markers, but did express inhibitory markers, indicating T cell exhaustion/inactivity [58]. T cell activation is also likely impaired as indicated by the variable MHCII expression on B cells. Th1 cells commonly classified as anti-tumorigenic were absent in this model. In the myeloid compartment, CD80, a known immunosuppressive marker, was expressed and this protein is associated with tumor progression and immune tolerance in ovarian cancer [59]. Given that M2 polarized macrophages are the dominant type in metastases of patients with HGSOC [60], we were surprised to find similar M1 and M2 numbers in the metastases of our model. Whereas in the primary tumor of our model, there was a bias towards M1 polarized macrophages [61]. However, M1 and M2 macrophages exist on a continuous spectrum, and standard gating protocols may not be optimal for ovarian cancer as macrophages with markers for both have been observed in patients [62]. We were technically limited in our efforts to probe NK cell biology in this study. It is important to note that all of our immune profiling, except for the immunotherapy studies, was performed on material collected from the ovarian intrabursal model. Although surgical injection may cause a localized inflammatory and immune response, the effects of this acute injury are likely mitigated by over the long time course of this model.

Finally, our immunotherapy studies indicated that this model is poorly responsive to immunotherapy. Given how this mirrors the clinical picture in ovarian cancer, this model will have substantial utility in understanding and enhancing response to current and future immunotherapies.

## 5. Conclusions

The development of the STOSE.M1 luc model offers another pre-clinical model for conducting translational ovarian cancer research. This model grows in immunocompetent mice, and the luciferase label enables tracking in vivo and quantitation of tumor growth. By developing and characterizing this model, we can identify mechanisms responsible for poor immunotherapy response in patients, test new immunotherapies, and identify immunotherapy re-sensitization strategies.

## Figures and Tables

**Figure 1 cancers-14-04219-f001:**
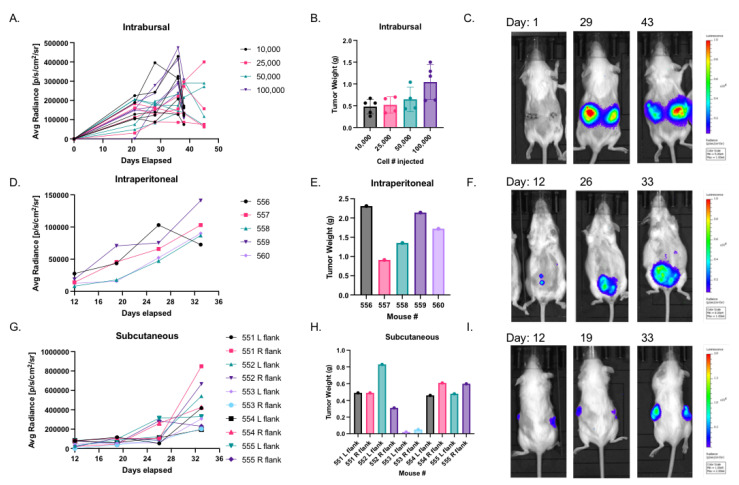
In vivo growth characteristics of luciferase labeled STOSE murine ovarian cancer cells: (**A**) Syngeneic ovarian intrabursal tumor growth of STOSE.M1 luc cells (1 × 10^4^–1 × 10^5^) in female FVB/NJ mice (*n* = 4–5 per group) measured by IVIS Lumina XR. (**B**) Weights of STOSE.M1 luc tumors, resulting from experiments in 1A. (**C**) Representative images of intrabursal tumor growth over course of experiment in 1A. (**D**) Syngeneic intraperitoneal tumor growth of STOSE.M1 luc (1 × 10^6^) cells in female FVB/NJ mice (*n* = 5, mouse # 556–560) measured by IVIS Lumina XR. (**E**) Total weight of intraperitoneal STOSE.M1 luc tumors per mouse, resulting from experiments in 1D. (**F**) Representative images of intraperitoneal tumor growth over course of experiment in 1D. (**G**) Syngeneic subcutaneous tumor growth of STOSE.M1 luc (1 × 10^6^) cells in left (L) and right (R) flank of female FVB/NJ mice (*n* = 5, mouse # 551–555) measured by IVIS Lumina XR. (**H**) Weights of STOSE.M1 luc tumors, resulting from experiments in 1G. (**I**) Representative images of subcutaneous tumor growth over course of experiment in 1G. Quantitative analysis of Avg Radiance (Figure 1A,D,G) refers to average radiance in ROI drawn around the individual tumor of each mouse.

**Figure 2 cancers-14-04219-f002:**
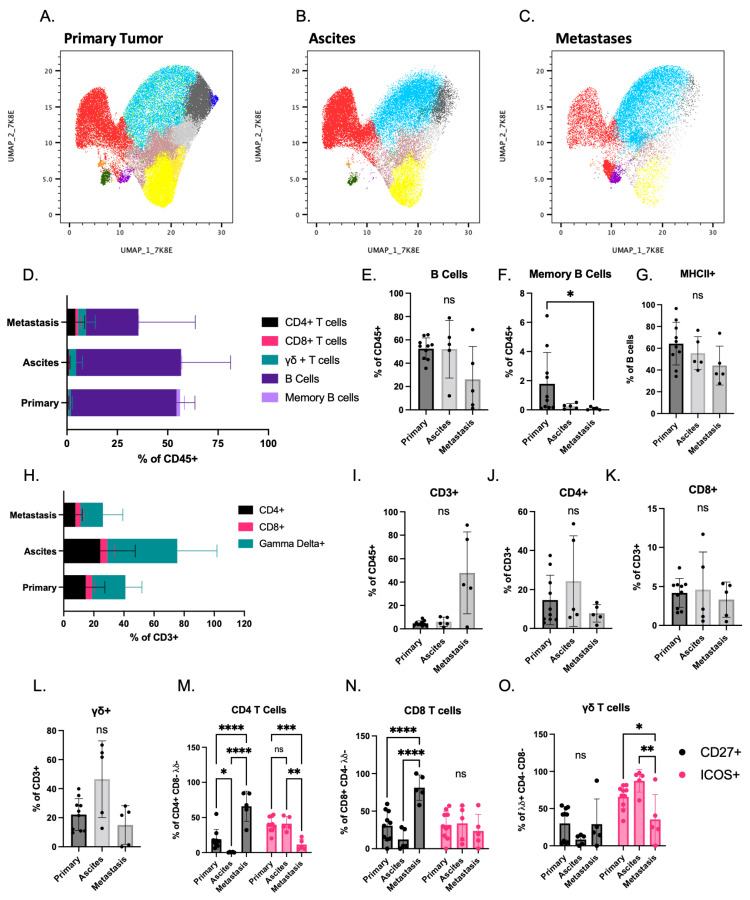
Lymphoid cell profiling of immune cells isolated from different sites in the STOSE.M1 luc murine ovarian intrabursal cancer model: (**A**–**C**) Uniform manifold approximation and projection (UMAP) plots overlayed with FlowSOM (Flow self-organizing map) of tumor infiltrating immune cells (CD45^+^) isolated from STOSE.M1 luc ovarian intrabursal model, propagated in FVB/NJ mice. Clustering was based on expression profiles of lymphoid cell surface markers. (**A**) Primary ovarian tumors (*n* = 10). (**B**) Ascites (*n* = 5). (**C**) Metastases (*n* = 5). Each dot represents an individual cell. (**D**) Percentages of T and B cells in each site of STOSE.M1 luc model. (**E**) Percentage of B cells (CD3${}^{-}$CD19^+^B220^+^). (**F**) Percentage of Memory B cells (CD3${}^{-}$CD19^+^B220^+^CD27^+^). (**G**) Percentage of MHCII^+^ B cells. (**H**) Percentage of T cells subsets in each site of STOSE.M1 luc model. (**I**) Percentage of CD3^+^ cells. (**J**) Percentage of CD4^+^ T cells. (**K**) Percentage of CD8^+^ T cells. (**L**) Percentage of γδ^+^ T cells. (**M**–**O**) Functional markers CD27 and ICOS in T cell subsets. Data are presented as mean ± standard deviation. Significance was calculated by Kruskal–Wallis test (**E**–**G**,**I**–**L**) followed by Dunn’s multiple comparisons test or two-way ANOVA (**M**–**O**) followed by Tukey’s multiple comparisons test. * *p* < 0.05, ** *p* < 0.01 and *** *p* < 0.001, **** *p* < 0.0001.

**Figure 3 cancers-14-04219-f003:**
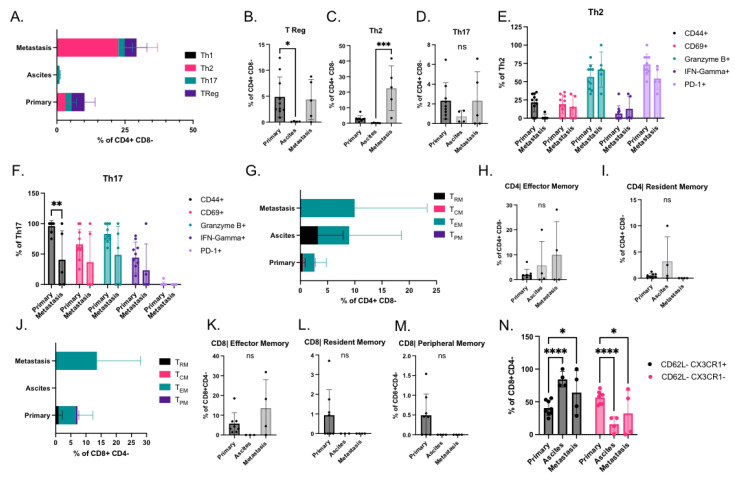
Profiling of Helper, Regulatory, and Memory T cell subsets in immune cells isolated from the different sites in STOSE.M1 luc murine ovarian intrabursal cancer model: (**A**) T Helper (Th) and T regulatory (T Reg) subsets as a % of CD4^+^ T cells in each site of STOSE.M1 luc ovarian intrabursal model, propagated in FVB/NJ mice. (**B**) Percentage of T Reg ( CD25^+^ FoxP3^+^) cells. (**C**) Percentage of Th2 cells (GATA3^+^). (**D**) Percentage of Th17 cells (RORγT^+^). (**E**,**F**) Functional markers in T helper subsets. (**E**) Th2. (**F**) Th17. (**G**) CD4^+^ Memory T cell subsets in each site of STOSE.M1 luc model. (**H**) Percentage of CD4^+^ Effector Memory T cells (CCR7(lo)/CD62L(lo)/CX3CR1(hi)/CD27(lo)/CD127(hi)). (**I**) Percentage of CD4^+^ Resident memory T cells (CCR7(lo)/CD62L(lo)/CX3CR1(lo/int)/CD44(hi)/CD127(hi)/CD103(hi)). (**J**) CD8^+^ Memory T cell subsets in each site of STOSE.M1 luc ovarian intrabursal model. (**K**) Percentage of CD8^+^ Effector Memory T cells (CCR7(lo)/CD62L(lo)/CX3CR1(hi)/CD27(lo)/CD127(hi). (**L**) Percentage of CD8^+^ resident memory T cells (CCR7(lo)/CD62L(lo)/CX3CR1(lo/int)/CD44(hi)/CD127(hi)/CD103(hi)). (**M**) Percentage of CD8^+^ Peripheral Memory T cells (CCR7(^+^/−)/CD62L(^+^/−)/ Cx3CR1(int)/CD27 (hi) /CD127 (hi)). (**N**) CD62L and CX3CR1 expression of CD8 memory populations. Data are presented as mean ± standard deviation. Significance was calculated by Kruskal–Wallis test (**B**–**D**,**H**,**I**,**K**–**M**) followed by Dunn’s multiple comparisons test or two-way ANOVA (**E**,**F**,**N**) followed by Tukey’s multiple comparisons test. * *p* < 0.05, ** *p* < 0.01 and *** *p* < 0.001, **** *p* < 0.0001.

**Figure 4 cancers-14-04219-f004:**
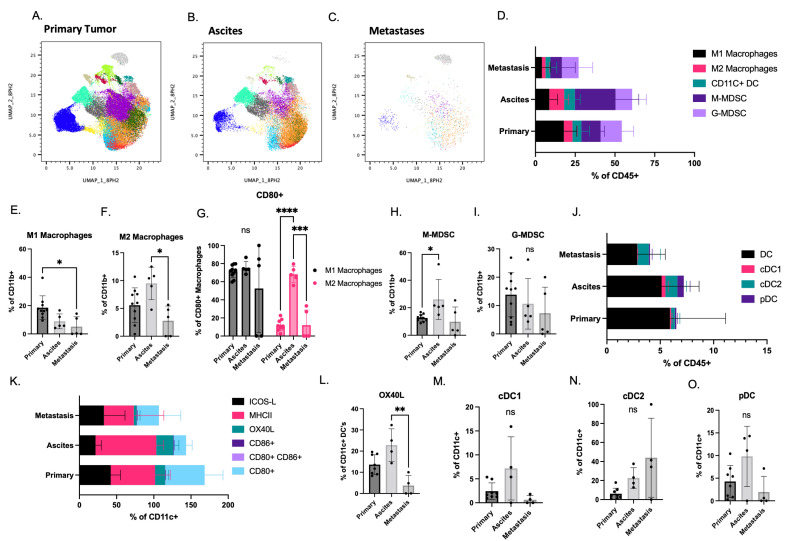
Myeloid cell profiling of immune cells isolated from different sites in the STOSE.M1 luc murine ovarian intrabursal cancer model: (**A**–**C**) UMAP plots overlayed with FlowSOM (Flow self-organizing map) of tumor infiltrating immune cells (CD45^+^) isolated from STOSE.M1 luc ovarian intrabursal model, propagated in FVB/NJ mice. Clustering was based on expression profiles of myeloid cell surface markers. (**A**) Primary ovarian tumors (*n* = 10). (**B**) Ascites (*n* = 5). (**C**) Metastases (*n* = 5). Each dot represents an individual cell. (**D**) Myeloid cells as a % of CD45^+^ immune cells in each site of STOSE.M1 luc model. (**E**) Percentage of M1 macrophages (CD206^+^ MHCII^+^). (**F**) Percentage of M2 macrophages (CD206^+^ MHCII${}^{-}$/low). (**G**) Frequency of CD80^+^ M1 and M2 macrophages. (**H**) Percentage of monocytic myeloid derived suppressor cells (Ly6C^+^ Ly6G${}^{-}$). (**I**) Percentage of granulocytic MDSCs (Ly6G^+^ Ly6C^+^). (**J**) Dendritic cells (DCs) as a % of CD45^+^ immune cells in each site of STOSE.M1 luc model. (**K**) Percentage of functional markers in CD11c^+^ DCs in each site of STOSE.M1 luc model. (**L**) Percentage of OX40L^+^ DCs. (**M**) Percentage of cDC1 cells (CD103^+^ CD11b${}^{-}$ B220${}^{-}$). (**N**) Percentage of cDC2 cells (CD11b^+^ CD103${}^{-}$ B220${}^{-}$). (**O**) Percentage of plasmacytoid DCs (B220^+^ CD103${}^{-}$ CD11b${}^{-}$). Data are presented as mean ± standard deviation. Significance was calculated by Kruskal–Wallis test (**E**–**G**,**N**,**P**–**R**) followed by Dunn’s multiple comparisons test or two-way ANOVA (**H**–**K**) followed by Tukey’s multiple comparisons test. * *p* < 0.05, ** *p* < 0.01 , *** *p* < 0.001 , **** *p* < 0.0001.

**Figure 5 cancers-14-04219-f005:**
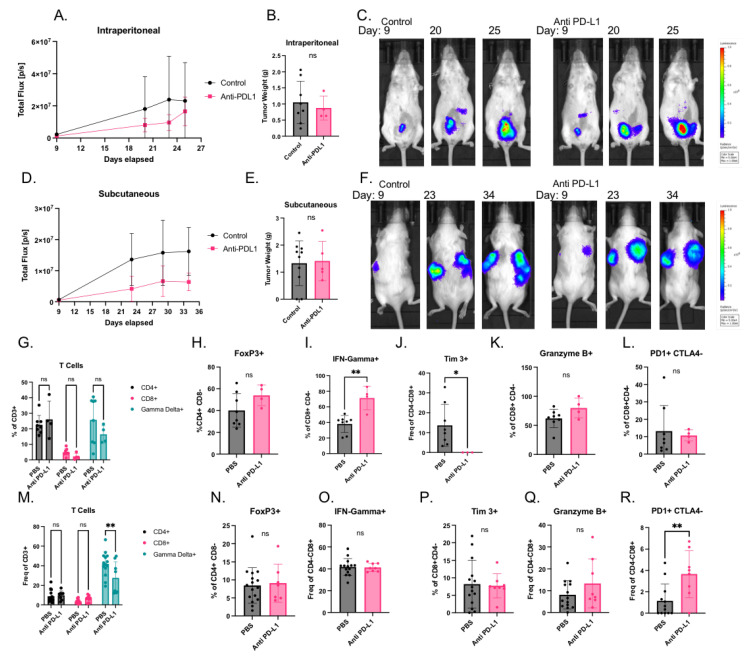
Response of subcutaneous and intraperitoneal STOSE.M1 luc models to immunotherapy, anti PD-L1: (**A**) Syngeneic intraperitoneal tumor growth of STOSE.M1 luc (1 × 10^6^) cells in female FVB/NJ mice (*n* = 4–8 per group) treated with control (PBS) or anti PD-L1 measured by IVIS Lumina XR . (**B**) Total weights of intraperitoneal STOSE.M1 luc tumors, resulting from experiments in 5A. (**C**) Representative images of intraperitoneal tumor growth over course of experiment in 5A. (**D**) Syngeneic subcutaneous tumor growth of STOSE.M1 luc (1 × 10^6^ per flank) cells in female FVB/NJ mice (*n* = 8–14 tumors per group) treated with control (PBS) or anti PD-L1 measured by IVIS Lumina XR . (**E**) Sum total weight of both subcutaneous flank STOSE.M1 luc tumors per mouse, resulting from experiments in 5D. (**F**) Representative images of subcutaneous tumor growth over course of experiment in 5D. (**G**–**L**) Immune cell profiling of intraperitoneal tumors treated with PBS or anti PD-L1 from 5A. (**G**) Percentage of T cells subsets in each treatment group. (**H**) Percentage of T Regulatory (FoxP3^+^ CD4^+^) cells. (**I**) Percentage of IFN-Gamma^+^ CD8^+^ T cells. (**J**) Percentage of Tim3^+^ CD8^+^ T cells. (**K**) Percentage of Granzyme B^+^ CD8^+^ T cells. (**L**) Percentage of PD1^+^ CTLA4${}^{-}$ CD8^+^ T cells. (**M**–**R**) Immune cell profiling of subcutaneous tumors treated with PBS or anti PD-L1 from 5D. (**M**) Percentage of T cells subsets in each treatment group. (**N**) Percentage of T Regulatory (FOXP3^+^ CD4^+^) cells. (**O**) Percentage of IFN-Gamma^+^ CD8^+^ T cells. (**P**) Percentage of Tim3^+^ CD8^+^ T cells. (**Q**) Percentage of Granzyme B^+^ CD8^+^ T cells. (**R**) Percentage of PD1^+^ CTLA4${}^{-}$ CD8^+^ T Data are presented as mean ± standard deviation. Significance was calculated by Mann–Whitney test (**B**,**E**,**H**–**L**), unpaired T-test (**N**–**R**) or two-way ANOVA followed by Šidák’s Multiple Comparisons test (**G**,**M**). ns, *p* > 0.05 * *p* < 0.05, ** *p* < 0.01.

**Table 1 cancers-14-04219-t001:** Characteristics of the Ovarian Cancer Models that were Developed.

Model	Injection Site	Cell # Injected	# of Mice Developing Tumors	# of Days to Tumor Formation	Ascites Volume (mL)	Metastases Formation
STOSE-luc	Ovarian Intrabursal	1 × 10^5^ to 1 × 10^6^	3/10	26	0	No
STOSE.M1 luc	Ovarian Intrabursal	1 × 10^4^ to 1 × 10^5^	9/10	21	3–9	Yes ^1^
	Intraperitoneal	1 × 10^6^	5/5	12	1–3	Yes ^2^
	Subcutaneous	1 × 10^6^	5/5	12	0–5	Yes ^3^
STOSE.M2 luc	Not tested in vivo	NA	NA	NA	NA	NA

^1^ Subcutaneous, peritoneum, ovary, adnexa, uterus, bowel, liver, ureter, mesentery, omentum, spleen, kidney, diaphragm, lungs. ^2^ Subcutaneous, pelvis, peritoneum, omentum, bowel, mesentery, adnexa. ^3^ Bilateral subcutaneous, ovary, omentum, pelvis, peritoneum, spleen.

## Data Availability

The data presented in this study are available on request from the corresponding author.

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
