# Peer review of "Development and Characterization of a Luciferase Labeled, Syngeneic Murine Model of Ovarian Cancer"

_cancers, 2022, doi:10.3390/cancers14174219_

Round 1

Reviewer 1 Report

Comments to Author:

The manuscript entitled “Development and characterization of a luciferase labeled, syngeneic murine model of ovarian cancer” by Shonagh et al., shared some interesting findings relevant to the field of ovarian cancer biology. Experiments are well designed and supported by the results and conclusions. I have some comments which need to be addressed to further improve the strength of the study as detailed below:

Comments

·     Authors are suggested to include a table for all types of models developed. The table should include information like no. of cells injected, days after the first tumor elapse, ascites formation, and metastatic sites.

Can authors explain why the timeline for PBS injection to control group is different from the anti-PDL1 group?   

Authors mentioned in the discussion section that all immune profiling was done in the ovarian intrabursal model only. This should be mentioned clearly in each legend section. Also, can authors provide justification for why immune profiling of the other two intraperitoneal and sub-cutaneous models was not done?

 Authors should discuss why only intraperitoneal and subcutaneous models were chosen for the immunotherapy study.

Reviewer 2 Report

Dear Authors,

This type of study has relevance in the field of ovarian cancer importantly pointing the need for models where cancer cell and cancer progression could be tracked and distinguished from the other cells types. Manuscript is well articulated however some information is missing or can be improved to highlight the work you have developed.

1. In line 250-252 you refer to organ site where metastasis can be found however no explanation is given in how the tissues were processed to get the cell suspensions and if the tissues were pooled. Every figure is labeled as metastasis without specifying about which tissue(s) are.

2. Supplementary figures show gating strategy for different cell population are missing legends where type of tissue is specified. As well gating strategy for each tissue analyzed is needed. Specially because low amount of cell population such as T cell (SF5) suggest comes from primary tumor but not for metastases site.

3. Using smooth plots for showing population is misleading when few cells can be found per gate therefore graphs of minor subsets are not suitable for conclusion unless percentages of cells are correlated with grams of tissue.

Round 2

Reviewer 1 Report

Recommended for publication